# Antispasmodic Effect of *Alstonia boonei* De Wild. and Its Constituents: Ex Vivo and In Silico Approaches

**DOI:** 10.3390/molecules28207069

**Published:** 2023-10-13

**Authors:** Opeyemi Josephine Akinmurele, Mubo Adeola Sonibare, Anthony A. Elujoba, Akingbolabo Daniel Ogunlakin, Oloruntoba Emmanuel Yeye, Gideon Ampoma Gyebi, Oluwafemi Adeleke Ojo, Abdullah R. Alanzi

**Affiliations:** 1Department of Pharmacognosy, Faculty of Pharmacy, Madonna University, Elele 512101, Nigeria; opeyemiakinmurele@gmail.com; 2Department of Pharmacognosy, Faculty of Pharmacy, University of Ibadan, Ibadan 200005, Nigeria; 3Comsat International Institute of Technology (CIIT), Abbotabad 22020, Pakistan; 4Department of Pharmacognosy, Faculty of Pharmacy, Obafemi Awolowo University, Ile-Ife 220101, Nigeria; tonyelu@yahoo.com; 5Phytomedicine, Molecular Toxicology, and Computational Biochemistry Research Laboratory (PMTCB-RL), Department of Biochemistry, Bowen University, Iwo 232101, Nigeria; oluwafemiadeleke08@gmail.com; 6H. E. J. Research Institute of Chemistry, International Center for Chemical and Biological Sciences, University of Karachi, Karachi 75270, Pakistan; bishopemmy2010@yahoo.com; 7Department of Chemistry, Faculty of Science, University of Ibadan, Ibadan 200005, Nigeria; 8Natural products and Structural (Bio-Chem)-Informatics Research Laboratory (NpsBC-RI), Department of Biochemistry, Bingham University, Karu 961105, Nigeria; gideonagyebi@gmail.com; 9Department of Pharmacognosy, College of Pharmacy, King Saud University, Riyadh 12271, Saudi Arabia; aralonazi@ksu.edu.sa

**Keywords:** medicinal plant, *Alstonia boonei*, antispasmodic activity, boonein, β-amyrin

## Abstract

Background: *Alstonia boonei*, belonging to the family Apocynaceae, is one of the best-known medicinal plants in Africa and Asia. Stem back preparations are traditionally used as muscle relaxants. This study investigated the antispasmodic properties of *Alstonia boonei* Stem back and its constituents. Method: The freeze-dried aqueous Stem back extract of *A. boonei*, as well as dichloromethane (DCM), ethyl acetate, and aqueous fractions, were evaluated for their antispasmodic effect via the ex vivo method. Two compounds were isolated from the DCM fraction using chromatographic techniques, and their antispasmodic activity was evaluated. An in silico study was conducted by evaluating the interaction of isolated compounds with human PPARgamma-LBD and human carbonic anhydrase isozyme. Results: The Stem back crude extract, DCM, ethyl acetate, and aqueous fractions showed antispasmodic activity on high-potassium-induced (K^+^ 80 mM) contractions on isolated rat ileum with IC_50_ values of 0.03 ± 0.20, 0.02 ± 0.05, 0.03 ± 0.14, and 0.90 ± 0.06 mg/mL, respectively. The isolated compounds from the DCM fraction were β-amyrin and boonein, with only boonein exhibiting antispasmodic activity on both high-potassium-induced (IC_50_ = 0.09 ± 0.01 µg/mL) and spontaneous (0.29 ± 0.05 µg/mL) contractions. However, β-amyrin had a stronger interaction with the two proteins during the simulation. Conclusion: The isolated compounds boonein and β-amyrin could serve as starting materials for the development of antispasmodic drugs.

## 1. Introduction

One of the most common diseases in humans is gastrointestinal (GI) disorder [1,2,3,4]. A spasm is defined as an uncontrollable muscular contraction of the smooth muscles or a transitory constriction of a passage such as the intestine, bladder, lungs, or stomach that is usually accompanied by pain and can be induced by endogenous acetylcholine [5]. Spasms are the root cause of the majority of GI disorders, such as colic, abdominal pain, flatulence, diarrhea, cramps, and irritable bowel syndrome. Smooth muscle spasms are caused by antagonistic muscarinic actions mediated by M3 receptors [6]. Spasms affect homeostasis in living organisms [7]. The control of lipid and glucose absorption is greatly influenced by the peroxisome proliferator-activated receptor (PPARγ receptor) [8,9]. The large intestine expresses PPARγ [10,11]. Its function in intestinal disorders, affecting the absorption of nutrients, is becoming more apparent [12,13]. Interest in the role of this receptor in the regulation of gut homeostasis has increased as a result of the discovery that it is the primary functional receptor mediating the aminosalicylate activities in inflammatory bowel diseases (IBDs), with potential implications for newer therapeutic targets [14,15,16]. At physiological pH, carbonic anhydrases (CAs) catalyze the conversion of CO_2_ to bicarbonate [17,18,19]. They participate in a wide range of biochemical and metabolic functions, including respiration, the biosynthesis of different metabolites (urea, glucose, fatty acids, and carbamoyl phosphate), the secretion of electrolytes, and the absorption of salt and water in the intestine [20,21,22]. When the secretion of these two enzymes is abnormal, it can lead to spasms or diarrhea in humans [23,24,25,26].

Antispasmodic agents, which relax and subdue muscle spasms or contractions, can counteract antagonistic muscarinic actions. They prevent stomach and intestine spasms by inhibiting cholinergic nerve impulses by blocking the action of acetylcholine neurotransmitters in the parasympathetic nervous system [4]. Antispasmodics are divided into two categories: anticholinergic (dicyclomine and hyoscyamine) and musculotropic (mebeverine). Each of them, however, has a variety of side effects, including dry mouth, narrow-angle glaucoma, tachycardia, and gastrointestinal tract obstructive disease [27].

Plant-derived substances, such as tropane alkaloids (atropine, hyoscine, scopolamine, and hyoscyamine), opium alkaloids (paperverine, codeine, and morphine), flavonoids (luteolin, quercetin, rutin, apigenin, and kaemferol), and essential oils (from caraway and peppermint), are thought to be safer and more cost-effective treatments [28]. Medicinal plants contain a high concentration of phytochemical constituents with antispasmodic properties, which help to relieve GI pain and contractions. Some antispasmodic medications of herbal origin are already used in conventional medicine. They play a critical role in the treatment of gastrointestinal motility disorders [29]. They are useful for relieving or calming colic caused by gastrointestinal muscle spasms as well as diarrhea caused by gastrointestinal hypermotility. For instance, paperverine from *Papaver somniverum* L. (Papaveraceae) is well-known for treating colic, whereas atropine from *Atropa belladonna* L. (Solanaceae) is an antimuscarinic agent used to treat gastrointestinal spasms caused by acetylcholine [30]. 

The Apocynaceae family, which includes the genus Alstonia, is widely dispersed in the tropics of Asia and Africa. Many studies have been conducted on the phytochemical components of *Alstonia* sp.; almost 400 chemicals have been identified [31,32,33]. The leaves of the species in this genus have long been employed in “Dai” ethnopharmacy to treat chronic respiratory disorders. These are well-known plants in Chinese medicine. Hospitals prescribe their leaf extract, which is a traditional Chinese medication with commercial value, and drug stores sell it over the counter [32,34]. *Alstonia boonei* is referred to locally by the Yoruba tribe in Nigeria as “Ahun” and is one of the numerous medicinal herbs used in recipes to treat malaria [35]. It is native to Africa and belongs to the Apocynaceae family, which contains 50 species that are widely dispersed across the continents of Africa, Asia, and America. Stem back preparations are used traditionally to treat a variety of conditions, including arterial hypertension, arthritis, cataracts, placenta retention, rheumatic pains, snakebites, venereal infections, malaria, measles, boils, wounds, and muscle relaxation [36,37,38]. This plant should not be used by women who are pregnant, nursing mothers, or who have liver dysfunction [39]. Due to their capacity to prevent parasitic nematodes from producing glutathione S-transferase, extracts of *A. boonei* may have anthelmintic effects [40]. According to Olajide et al. [36], *A. boonei* stem back has anti-inflammatory, antipyretic, and analgesic properties. This study investigated the antispasmodic effect of *Alstonia boonei* Stem back and its constituents via ex vivo and in silico approaches.

## 2. Results

### 2.1. Antispasmodic Effect of Crude Extract and Fractions of Alstonia boonei

The Stem back crude extract, DCM, ethyl acetate, and aqueous fractions showed antispasmodic activity on high-potassium-induced (K^+^, 80 mM) contractions on isolated rat ileum with IC_50_ values of 0.03 ± 0.20, 0.02 ± 0.05, 0.03 ± 0.14, and 0.90 ± 0.06 mg/mL, respectively. Their effect on spontaneous contractions of the isolated rat ileum is presented in Table 1. The DCM fraction had the highest antispasmodic effect on the rat ileum. Out of all the fractions, DCM showed the highest relaxation activity (Figure 1, Figure 2, Figure 3 and Figure 4) on both spontaneous and high-potassium-induced contractions on isolated rat ileum, with IC_50_ values of 0.31 ± 0.02 and 0.02 ± 0.05 mg/mL, respectively.

### 2.2. Structural Elucidation of Compounds ***1*** and ***2***

Compound **1** was obtained as a white powder, and when viewed under UV at the 254 nm and 365 nm wavelengths, it gave no fluorescence (due to the absence of a conjugated bond in the compound), but it gave an orange coloration with 10% sulfuric acid at 100 °C. Three markers were contained in the compound; these include a carbonyl group at C-1, a methyl group at C-8, and a hydroxyl group at C-7. The proton NMR showed the existence of fourteen protons in total: a hydroxyl group at *δ_H_* 6.13 ppm, which integrated as a singlet with a broadband and attached to C-7; a methyl proton (C-8) at *δ_H_* 1.47 ppm, which integrated as a doublet; three methylene protons at *δ_H_* 4.21 ppm (which integrated as multiplets), at *δ_H_* 1.84 ppm (which integrated as multiplets), and at *δ_H_* 1.34 ppm (found at positions 3, 4, and 6, respectively); as well as four methine protons at *δ_H_* 2.92 ppm (dd), *δ_H_* 4.21 ppm (C-7), *δ_H_* 2.28 ppm, and *δ_H_* 2.92 ppm. The protons on C-3 and the hydroxyl group appeared at *δ_H_* 4.21 and 4.22 ppm, respectively. The ^13^C-NMR spectrum showed one carbonyl group at *δ_C_* 178.9 ppm. The broadband and DEPT spectra showed the existence of nine carbon signals, including a quaternary carbon, four methine carbons, three methylene carbons, and a methyl group. The DEPT 90 spectrum showed the existence of four methine (CH) groups, while the DEPT 135 spectrum showed the existence of a methyl group (CH_3_) whose peak exists upfield (on the positive axis of the spectrum); four methine (CH) whose peaks are found downfield with the methyl group; and three methylene (CH_2_) signals located on the negative axis of the DEPT 135 spectrum, which was the opposite of the positioning of CH and CH_3_. The long-range HMBC correlation of H-3 at *δ_H_* 4.21 appeared with C-4 at *δ_C_* 29.8 and C-5 at *δ_C_* 34.3 ppm, while H-4 (*δ_H_* 1.84 ppm) showed a correlation with C-5 (*δ_C_* 34.3 ppm) and C-9 (*δ_C_* 47.9 ppm); H-8 at *δ_H_* 2.28 ppm revealed a correlation with C-7 and C-9 at *δ_C_* 74.4 and 47.9 ppm, respectively; H-9 showed a correlation with C-5, C-8, C-10, and C-1 at *δ_C_* 34.4, 44.9, 14.3, and 178.9 ppm, respectively; H-10 at *δ_H_* 1.47 resonated with C-5, C-7, C-8, and C-9 at *δ_C_* 34.3, 74.4, 44.9, and 47.9, respectively). The EIMS data (low and high resolution) of compound **1** suggested a molecular mass of 170.0 g/mol with the molecular formula C_9_H_14_O_3_. Thus, Compound **1** was elucidated as (4As, 6S, 7R, 7aS)-6-hydroxy-7-methyl-4,4a,5,6,7,7a-hexahydro-3H-cyclopenta[c]pyran-1-one (boonein), whose spectral data were comparable with those previously reported for boonein by Marini-Bettolo et al. [41]. All the proton and carbon signals were apportioned based on 1H-1H COSY, DEPT analysis, HMQC, and HMBC. The chemical structure of compound **1** is shown in Figure 5. 

Compound **2** was obtained as a needle-like crystal from DCM: a Hex (80:20) fraction of column chromatography that did not fluoresce under UV light at both a short and a long wavelength (254 and 365 nm) but when sprayed with chromogenic reagents (ceric sulfate and 10% sulfuric acid) gave a purple color. Five out of the six double-bond equivalents were adjusted in a pentacyclic carbon framework; one manifested as a C=C double bond. The ^1^HNMR spectrum revealed the existence of eight methyl singlets at C_23_-C_30_ with *δ_C_* values of 15.7, 28.5, 16.5, 17.0, 26.1, 28.6, 33.4, and 30.0 ppm, including an olefinic proton at *δ_H_* 5.16 ppm and an oxygenated proton at *δ_H_* 3.12 ppm, further suggesting an oleanane-type triterpenoid nucleus. All the proton and carbon signals were positioned based on ^1^H-^1^HCOSY, DEPT analysis, HMQC, and HMBC. The EIMS data (low and high resolution) gave the molecular mass as 426.8 gmol^−1^, which corresponds with the molecular formula C_30_H_50_O. Thus, Compound **2** was elucidated as 3b-hydroxylolean-12-ene, whose spectral data were comparable with those previously reported and identified as β-amyrin by Okoye et al. [42]. The chemical structure of compound **2** is shown in Figure 6.

### 2.3. Antispasmodic Studies on Compounds ***1*** and ***2***

Figure 7 and Figure 8 show the antispasmodic activities of compounds **1** and **2** isolated from the DCM fraction of *A. boonei* Stem back on spontaneous and high-potassium-induced (80 mM) contractions on isolated rat ileum. Compound **1** shows concentration-dependent antispasmodic activity at concentrations 0.003–10.0 µg/mL on spontaneous contractions with an IC_50_ value of 0.29 ± 0.05, while Compound **2** shows spasmodic activity at concentrations 0.003–5 µg/mL and also an antispasmodic effect at 10 µg/mL on spontaneous contractions with an IC_50_ value of 2.2 ± 0.7. Table 2 presents the IC_50_ values of the antispasmodic activities of compounds **1** and **2** isolated from the DCM fraction of *A. boonei* Stem back on spontaneous and high-potassium-induced (80 mM) contractions on isolated rat ileum.

### 2.4. Molecular Docking

The results from the molecular docking analysis show that β-amyrin, boonein, and the reference compounds recorded binding energies of −9.4, −6.4, and −9.4 Kcal/mol, respectively, for human carbonic anhydrase isozyme 1 and of −8.4, −6.0, and −8.2 Kcal/mol, respectively, for the human PPARgamma-LBD protein. It was observed that β-amyrin demonstrated the highest binding tendencies to both proteins, which were comparable with the reference inhibitors. Although boonein was docked with a lower binding energy compared to β-amyrin and the reference inhibitors, both β-amyrin and boonein were docked into the binding site in an orientation close to that of the reference compounds (Figure 9 and Figure 10). The amino acid interaction analysis of both compounds with the protein targets revealed that, although boonein was docked with a lower binding energy to both proteins, it interacted with both the hydrogen bonds and hydrophobic interactions with both proteins, while β-amyrin interacted only with the hydrophobic interactions with the binding site residues (Table 3). The binding free energy of the compounds to both the 4F9M and 5E2M systems shows that β-amyrin had a higher binding free energy (−27.02 and −12.88 Kcal/mol, respectively) than boonein (−0.86 and −4.19 Kcal/mol, respectively) (Table 4). After the cluster analysis, the amino acids for each protein were renumbered to start from number 1 until the end of the sequence. All gaps within the sequence were removed, and this was different from the number in the retrieved crystal structures (Figure 11). From these clusters, representative structures were selected for further analysis. The results further show that the interactions with important catalytic residues were preserved during the simulations (Table 5).

## 3. Discussion

Crude extract and fractions of *Alstonia boonei* Stem back s have been shown to exhibit ex vivo antispasmodic potential; however, the DCM fraction has shown greater effectiveness. The solvent-partitioned fractions displayed antispasmodic and spasmodic properties. The organic components, particularly the less polar fractions, were where the antispasmodic activity was concentrated. Makrane et al. [43] found a similar outcome for *Origanum majorana* L. This was also supported by a previous study by Bashir et al. [44] on the DCM fraction of *Calendula officinalis* in both spontaneous and high-potassium-induced contractions of isolated rabbit jejunum. In this study, the DCM fraction demonstrated highly significant antispasmodic activities on both spontaneous and high-potassium-induced contractions in the isolated rat ileum. The aqueous extract of *A. boonei* Stem back was reported to have antispasmodic and spasmodic effects, and the same was reported for the aqueous fraction of *Calendula officinalis*. The ethyl acetate fraction of *Calendula officinalis* also exhibited both antispasmodic and spasmodic activities at different concentrations [44]. The observed spasmodic effect of the polar elements in this partitioned fraction may be responsible for the reported abortifacient property of the species in the genus Alstonia [45]. 

The effect of the *A. boonei* stem back DCM fraction on spontaneous movements of the rat ileum observed in this study could be due to interference with the Ca^2+^ influx through Voltage-Dependent Calcium Channels (VDCs) from the intercellular medium or Ca^2+^ release from the sarcoplasmic reticulum [46]. The tissue was pre-treated with 80 mM [K^+^] to validate the interaction of the isolated compounds with VDC. As a result of the membrane depolarization, the VDC opened, allowing Ca^2+^ to penetrate the cytoplasm. Any substance that inhibits KCl-induced contractions is referred to as a VDC blocker [47], so crude extract and all fractions are VDC blockers.

Two bioactive compounds of *A. boonei* stem back aqueous crude extract were isolated in the DCM fraction. Compound **1** demonstrated greater antispasmodic activity than compound **2** in the current study. Despite this, the two compounds had biological activities on the smooth muscles of the gastrointestinal tract. Compound **1** may be an active antidiarrheal agent based on its high antispasmodic effect on the isolated rat ileum. Such a phytochemical venture could result in a single-compound formulation, similar to what is seen in conventional pharmaceutical or orthodox therapy practices. A compound formulation has many advantages, including the elimination of unwanted additional constituents, the ability to use different dosage forms (including parenteral solutions), the ease of analysis for quality control, the ease of monograph compilation for the finished product, the monitoring of deterioration in storage, etc. Compound **1** significantly reduced the contraction of the isolated rat ileum caused by high [K^+^]. It was ten times more potent than Compound **2**, which only had a partial relaxation effect on the isolated rat ileum. It could be one of the synergistic constituents with echitamidine previously reported in *A. boonei* as an antispasmodic in the ileum of guinea pigs [48]. Compound **2**, on the other hand, exhibited a biphasic property, acting as both a spasmodic and an antispasmodic. The isolation of these compounds can thus aid in the identification of the specific constituent(s) responsible for a particular bioactivity, either alone or in collaboration with other constituents, as may be the case in the current study. Some alkaloids, such as benzylisoquinoline alkaloids [49], peracetylated penstemonoside, aucubin and catalpol [50], chelidonine, protopine, coptisine [51], and protobeberin [52], have also been reported to possess antispasmodic activity.

Compound **1** also acted as a VDC blocker. The inhibitory effect of isolated compounds on spontaneous and high-[K^+^]-induced contractions of the rat ileum appears to be due to a Calcium Channel Blocker (CCB) effect and may be responsible for these samples’ antidiarrheal effect. This indicates the presence of Ca^2+^ antagonists in the extracts and that Compound **1** isolated from the DCM fraction of *A. boonei* is one of many constituents of *A. boonei* extract containing Ca^2+^ antagonists, implying its utility in treating tissue hyperactivity in diarrhea. These findings may contribute to a better understanding of the antispasmodic mechanisms that reduce intestinal motility and can be used in non-infectious diarrhea.

Different parts of medicinal plants have yielded antispasmodic compounds. *Ipomoea pes-caprae* (L.) R. Br. produced the isoprenoids β-damascenone and *E*-phytol, which have antispasmodic properties [53]. *Allium cepa* L. var. Tropea yielded quercetin, quercetin 4^I^-glucoside, taxifolin, taxifolin 7-glucoside, and phenylalanine were discovered to have antispasmodic properties on the isolated ileum of a guinea pig [54]. Luteolin, acteoside, plantamajoside, and catalpol peracetate were the compounds isolated from *Plantago lanceolata* L. that inhibited the guinea-pig ileum’s ability to contract in response to ACh, unlike catalpol, isoacteoside, lavandulifolioside, or aucubin. The guinea-pig trachea’s barium-induced contractions were lessened by luteolin and acteoside [55]. Ileal and tracheal smooth muscle contractions were inhibited by moracin O, which was isolated from the root bark of *Morus nigra* [56].

In the MD simulation analysis, the structural integrity and stability of the bound structures are compared to the unbound structures of proteins through the various conformational fluctuations that occurred in the MD simulated environment [57]. In these studies, the RMSD, RMSF, SASA, RoG, and number of H-bonds were computed from the MD trajectories. The plots were presented as a function of time. The RMSD plots show the extent of the deviation of each frame from the initial structure and are hence used to assess the protein stability of the systems [58]. The RMSD of the two systems is represented in Appendix A. The 4F9M systems were equilibrated around 10 ns with average RMSD values of 1.721, 1.972, and 1.819 Å for the unbound enzyme, β-amyrin, and boonein complexes, respectively. With the 5E2M systems, the unbound protein and the 5E2M_β-amyrin complex were equilibrated around 10 ns, while the 5E2M_boonein complex was still fluctuating around 10 ns until the end of the simulation. The average RMSD values were 2.109, 2.419, and 3.630 Å for the unbound enzyme, β-amyrin, and boonein complexes, respectively. The RMSF plots show the flexibility of different regions of the protein [59]. There are often spikes that occur at the N and C terminal ends of the proteins, corresponding to the terminal motions. The mean RMSF values for the 4F9M systems are 0.9456 Å for the apoenzyme and 1.061 and 1.177 Å for the enzyme complexed with β-amyrin and boonein, respectively. The 5E2M systems had average RMSF values of 1.1035 for the apoenzyme and 1.7831 and 1.369 Å for the β-amyrin and boonein complexes, respectively. The RoG measures the compactness of the systems [60]. A stably folded protein structure presents a steady RoG plot. Figure 5 shows the RoG plots of the two protein systems. The plots for the 4F9M systems show a steady progression during the simulation period, but 5E2M_boonein displayed high fluctuations. The mean RoG values for the 4F9M systems are 19.192, 19.411, and 19.370 Å for the apoprotein-, β-amyrin-, and boonein-complexed systems, respectively. Those for the 5E2M are 17.963 for the apoenzyme and 18.116 and 18.449 Å for the enzyme complexed with β-amyrin and boonein, respectively. The SASA plots show the degree of solvent accessibility by the surface of the proteins [61]. After equilibration at the stage, the boonein complexes demonstrated a higher degree of fluctuation compared to the other system. The mean SASA values for the 4F9M systems are 14,677.31, 15,166.09, and 15,379.48 Å^2^ for the apoprotein-, β-amyrin-, and boonein-complexed systems, respectively. Those for the 5E2M are 13,252.86 for the apoenzyme and 13,499.4, 15,166.09, and 14,116.53 Å^2^ for the enzyme complexed with β-amyrin and boonein. The 4F9M protein system had an average number of hydrogen bonds of 62.37, 61.355, and 59.555 for the apoenzyme-, β-amyrin-, and boonein-complexed systems, respectively, while the 5E2M protein system had an average number of hydrogen bonds of 57.545, 57.11, and 5.178 for the apoenzyme-, β-amyrin-, and boonein-complexed systems, respectively. Although the analysis of the interaction between boonein and the two proteins revealed a higher number of hydrogen bonds that were formed with the proteins, the analysis from the MD simulation trajectories shows a lower average number of hydrogen bonds in both proteins. This may signify that the bonds were lost during the simulation. From all the analysis made from the trajectories of both complex systems, it can be suggested that the stronger interaction between β-amyrin and the two proteins was preserved during the simulation.

The quantitative free energy of the binding of ligands to proteins in a dynamic environment has proven to be a more reliable and accurate calculation for binding affinity [61]. In this study, the static docking calculations were further corroborated by the binding free energy calculation in the dynamic environment. The decomposition of binding free energy based on the contribution of amino acids further showed that boonein lost the majority of its interaction with both protein targets. Hence, there was no contributing amino acid that was recorded within 10 Å, and only the free energy contribution per residue for β-amyrin to both proteins was reported. The cluster analysis of the trajectories obtained from the MD simulation and interaction analysis in the representative conformers of the clusters reveals that the interactions (mostly hydrophobic interactions) between β-amyrin-bound complexes to both proteins that were identified from the static docking experiment were preserved at different times during the simulation.

## 4. Materials and Methods

### 4.1. Plant Material, Extraction, and Partitioning

*Alstonia boonei* De Wild. (Apocynaceae) stem back was collected in September 2015 behind the Department of Physiology at the University of Ibadan in Ibadan, Nigeria. Authentication of the fresh plant was carried out by Mr. Ifeoluwa Ogunlowo, the herbarium curator at the Ife Herbarium, Department of Botany, Obafemi Awolowo University, Ile-Ife, with voucher number FPI 2169. By rinsing the Stem back s of *Alstonia boonei* with tap water, lichens and dirt were removed. To expedite air-drying under shade for 14 days, the Stem back s were cut into pieces. The dried samples were ground into a coarse powder before being stored in an airtight container. A powdered sample of *A. boonei* Stem back (4 kg) was macerated in distilled water for 72 h before being filtered and concentrated in vacuo at 50 °C using a Rotavapor (Buchi, Germany), then freeze-dried using a freeze-drier (Gunman, Germany). The extract was then freeze-dried (Gunman, Germany) to obtain a crunchy, dried brownish extract, weighed, and refrigerated until needed. The extract’s percentage yield was calculated. The aqueous extract (100 g) of *Alstonia boonei* Stem back was partitioned into dichloromethane (DCM), ethyl acetate, and the residue, which was the aqueous fraction.

### 4.2. Ex Vivo Antispasmodic Assay 

#### 4.2.1. Experimental Animals for Antispasmodic Assay

Male Wistar albino rats (150–200 g) were used for this study. They were housed, fed a standard diet and water ad libitum, and kept at 23–25 °C at the Animal House. The animals had fasted for 18 h before the experiment with water ad libitum. This was carried out with prior approval of the Animal Use Ethics Committee of the Biochemistry Program, Bowen University, Iwo, Nigeria (BPBUI/08/22/01).

#### 4.2.2. Antispasmodic Effect of Crude Extract and Fractions

Experimental animals were euthanized by cervical dislocation. Antispasmodic activity of the crude extract and fractions was studied on the isolated ileum of mature albino Wistar rats as described by Gilani et al. [62]. The segmented ileum, measuring 2 cm in length, was suspended in a 10 mL tissue bath filled with Tyrode solution. The solution, composed of NaHCO_3_ (11.90 mM), MgCl_2_ (1.05 mM), KCl (2.68 mM), NaCl (136.9 mM), CaCl_2_ (1.8 mM), glucose (5.55 mM), and NaH_2_PO_4_ (0.42 mM), was bubbled with carbogen gas at 37 °C. A constant resting tension of 1 g was applied to the tissues (ileum) throughout the experiment. Isometric contractions were recorded using force displacement transducers connected to a Power Lab Data Acquisition System (AD Instruments, Sydney, Australia) attached to a computer installed with labchart software (version 6). Tissues were equilibrated for a minimum of 30 min and stabilized with a sub-maximal concentration of acetylcholine (0.3 μM), which was washed off immediately and replaced with Tyrode solution before the start of the experiment. The spontaneous rhythmic contractions exhibited by the rat ileum under the above experimental conditions gave room for the direct testing of the antispasmodic activity without the use of any agonist. For determination of the antispasmodic effects of the extract and fractions, dosing of test drugs was carried out cumulatively by serial dilution of 300–3 µg/mL at 0.01–10 µg/mL at 3–5 min intervals. A high-potassium-ion concentration (high [K^+^]) (80 mM) was applied to depolarize the preparations as described by Farre et al. [63]; this was performed to assess whether the antispasmodic effect of the extracts was mediated through Calcium Channel Blockade (CCB). The addition of high [K^+^] to the tissue bath resulted in a sustained contraction. Relaxation of the ileum pre-contracted with [K^+^] by the extract was expressed as a percentage of the control response mediated by high [K^+^]. The assay was conducted in triplicates for each test sample. This assay was repeated for the determination of the antispasmodic potential of the isolated compounds.

### 4.3. Isolation of Compounds ***1*** and ***2*** from A. boonei

Silica gel (150 g, 70–230 mesh sizes) was made into a slurry with 100% *n*-hexane and packed into a glass column (diameter = 40 mm). It was compactly packed after being eluted with 100% n-hexane. The column was loaded with five grams (5.0 g) of dichloromethane (DCM) fraction pre-adsorbed on silica gel (5 g). The loaded sample was eluted using a gradient elution method that began with *n*-hexane (100%), then moved on to *n*-hexane: DCM (90:10), DCM: ethyl acetate (90:10), and ethyl acetate: methanol (90:10). After achieving absolute methanol polarity, fractions were collected in 100 mL volume beakers, concentrated in vacuo, and transferred into small vials, yielding 150 fractions that were pooled into 10 fractions using TLC (I-X). Column chromatography was used to further purify fractions II and V using DCM: ethyl acetate (80:20) in increasing polarity until absolute ethyl acetate polarity was achieved for fraction II and *n*-hexane: ethyl acetate (80:20) for fraction V. Fractions II and V yielded compound **1** (80 mg, white powder) and compound **2** (150 mg, crystal), respectively.

### 4.4. Structure Elucidation of Compounds ***1*** and ***2***

To identify and elucidate the structures of the isolated compounds from the dichloromethane fraction of *A. boonei* Stem back, ^1^H NMR and ^13^C NMR using a Bruker Ascend 400 MHZ spectrometer (Bruker Instruments Incorporation, Billerica, MA, USA), mass spectroscopy (EI/mz), and other 2D-NMR spectroscopic techniques including COSY, NOESY, DEPT-90, and DEPT-135 were used. Deuterated pyridine was used to dissolve the compounds (C5D5N). Chemical shifts (∂) were measured in parts per million (ppm), coupling constants (J) in hertz, and trimethylsilane (TMS) as an internal standard. Using distortionless enhancement by polarization transfer (DEPT), carbons were distinguished and classified as methane (CH), methylene (CH_2_), or methyl (CH_3_) groups. Furthermore, heteronuclear single quantum coherence (HSQC) was used to directly correlate proton and carbon nuclei through one bond, and Heteronuclear Multiple Bond Correlation (HMBC) was used to obtain long-range correlations of proton and carbon nuclei through two, three, or four bonds. Mass spectroscopy (EI-MS) was used to validate the compounds’ molecular weight.

### 4.5. In Silico Studies 

#### 4.5.1. Retrieval and Preparation of Proteins

The 3D crystallographic structures of human PPARgamma-LBD complexed with a cercosporamide derivative (PDBID: 4F9M) and human carbonic anhydrase isozyme 1 complexed with 3-(cyclooctylamino)-2,5,6-trifluoro-4-[(2-hydroxyethyl)sulfonyl]benzenesulfonamide (PDBID: 5E2M) were retrieved from http://www.rcsb.org (Protein Data Bank, accessed on 2 September 2022). The co-crystallized ligand and water molecules accompanying the protein were deleted, while the hydrogen atoms that were missing were added using MGL-AutoDockTools (ADT, v1.5.6). Kollamn charges were added as the partial atomic charge [64]. This procedure was applied to both proteins. 

#### 4.5.2. Ligands Preparation 

The Structure Data Format (SDF) of boonein and β-amyrin was assessed from www.pubchem.ncbi.nlm.nih.gov (PubChem database, accessed on 2 September 2022). The SDF format was further converted to PDB format using the SaveAs option in Discovery Studio 2021. The PDB format of the compounds was imported into PyRx 0.8 using the incorporated Open Babel tool in the software. The conjugate gradient descent was applied as the optimization algorithm, while the Universal Force Field (UFF) was employed as the energy minimization parameter using the Open Babel tool [65]. The option convert AutoDock (PDBQT) was used to convert the ligands into PDBQT. The compounds were saved in the ligands section of the software. 

#### 4.5.3. Targeted Molecular Docking

Autodock Vina [66], incorporated into PyRx 0.8, was employed for docking and determination of binding energies of the two compounds and reference inhibitors to both proteins. The binding site of the protein targets was determined by mapping the area accommodated by the co-crystallized ligands. The grid boxes used for the docking studies were obtained by selecting the amino acid residues that interacted with the native ligands and drawing a grid box to enclose them. A cluster analysis based on RMSD values concerning the starting geometry was subsequently executed, and the minimum energy conformation of the most populated cluster was taken as the most reliable solution. The binding energies of the compounds for both protein targets were recorded. The docked positions and interactions with catalytic residues were further analyzed. The molecular interactions between the outstanding compound and the protein targets were viewed using Discovery Studio Visualizer, BIOVIA, 2021.

#### 4.5.4. Molecular Dynamics 

The unbound structures of 4F9M and 5E2M and the complexed structures with boonein and β-amyrin were further subjected to molecular dynamics (MD) simulation to evaluate the stability of the bound structure as compared to the unbound structures. The 100 ns simulation was performed on GROMACS 2019.2, employing the GROMOS96 43a1 forcefield on the WebGRO [67,68,69]. The topology files for the ligands were obtained from http://davapc1.bioch.dundee.ac.uk/cgi-bin/prodrg (PRODRG webserver, accessed on 2 September 2022) (Schüttelkopf and Van Aalten, 2004). The protein and the protein complexed with the ligands were solvated in a cubic box with a four-point (TIP4P) water model. A total of 0.154 M (physiological concentration) was set for neutralized NaCl ions, for which the periodic boundary conditions were employed. A 10,000-step minimization for each of the systems was executed using the steepest descent algorithm for 0.3 nanoseconds in the constant number of atoms, volume, and temperature (NVT) ensemble. An equilibration for 0.3 nanoseconds at a constant number of atoms, pressure, and temperature (NPT) was further applied. Pressure and temperature were set to 1 atm and 310 K, respectively, and maintained using a Parrinello–Rahman barostat and velocity rescale, respectively. Also, with a time step of 2 femtoseconds, the leap-frog integrator was employed. A total of 100 ns of the simulation was completed in each of the systems, with a total of 1000 frames obtained from a 0.1 nanosecond snapshot. The obtained MD trajectories were evaluated using the TK console scripts in VMD [70]. The number of H-bonds, SASA, RMSD, RoG, and RMSF parameters were analyzed.

#### 4.5.5. Binding Free Energy Calculation Using MM-GBSA

For an in-depth analysis of the interaction of boonein and β-amyrin with their respective protein targets, free energy of binding using Molecular Mechanics—Generalized Born Surface Area (MM-GBSA) computation was implemented. The gmx_MMPBSA package was used to compute the free energy of binding using the MM-GBSA algorithm for each complex system. The binding energies were decomposed to obtain the amino acids within 10 Å around the ligand [71,72]. The concentrations of salt and solvation (igb) were set to 0.154 M and 5, respectively, while the internal and external dielectric constants were set to 1.0 and 78.5, respectively. Other parameters were set as defaults. The MM-GBSA method is depicted in Equation (1).
(1)∆G=Gcomplex − Greceptor−Gligand

Different energy terms were calculated according to Equations (2)–(6).
(2)∆Gbinding=∆H−T∆S 
(3)∆H=∆Egas+∆Esol 
(4)∆Egas=∆Eele+∆EvdW
(5)∆Esolv=EGB+ESA
(6)ESA=γ·SASA
where ∆H is the calculated enthalpy from solvation-free energy (E_sol_) and gas-phase energy (E_gas_). So as to compare the relative binding free energies, T∆S, which represents the entropic contribution to the free binding energy, was not calculated in this study. E_gas_ comprises van der Waals (E_vdW_) and electrostatic (E_ele_) terms. E_sol_ was computed from the polar solvation energy (E_GB_), while (E_SA_), the nonpolar solvation energy, was assessed from the accessible solvent surface area [73,74].

#### 4.5.6. Clustering of Molecular Dynamic Trajectory

A further study was undertaken to investigate to what extent the interaction of the compounds with the catalytic residue was preserved during the 100 ns atomistic MD simulation analysis. The MD trajectories for the protein–ligand systems were subjected to cluster analysis using TTClust V 4.9.0, which employed the elbow method. After the conformation clustering, conformers that would serve as representative structures were chosen. The Protein–Ligand Interaction Profiler (PLIP) [75] was used for the interactive analysis. Unlike the downloaded crystallized structures with gaps in the numbering of amino acid residues, during the cluster analysis, the amino acid residues were automatically renumbered to start from number 1 until the last number of the protein sequence.

### 4.6. Statistical Analysis 

All results were expressed as mean ± SEM, and in the isolated tissue experiments, the data were analyzed by non-linear regression. Differences were considered statistically significant when the probability value was *p* < 0.05. GraphPad prism 5 software (GraphPad, San Diego, CA, USA) was used for the analysis.

## 5. Conclusions

From the DCM fraction with the highest antispasmodic activity, isolated compounds **1** and **2** relaxed the spontaneous and high-potassium-induced contractions of isolated rat ileum. Boonein (**1**) had a stronger antispasmodic action than β-amyrin (**2**), suggesting that it could be one of the major antidiarrheal agents in *A. boonei*. However, β-amyrin had a stronger interaction with the two proteins during the simulation. Therefore, the isolated compounds, boonein and β-amyrin, could serve as starting materials for the development of antidiarrheal drugs.

## Figures and Tables

**Figure 1 molecules-28-07069-f001:**
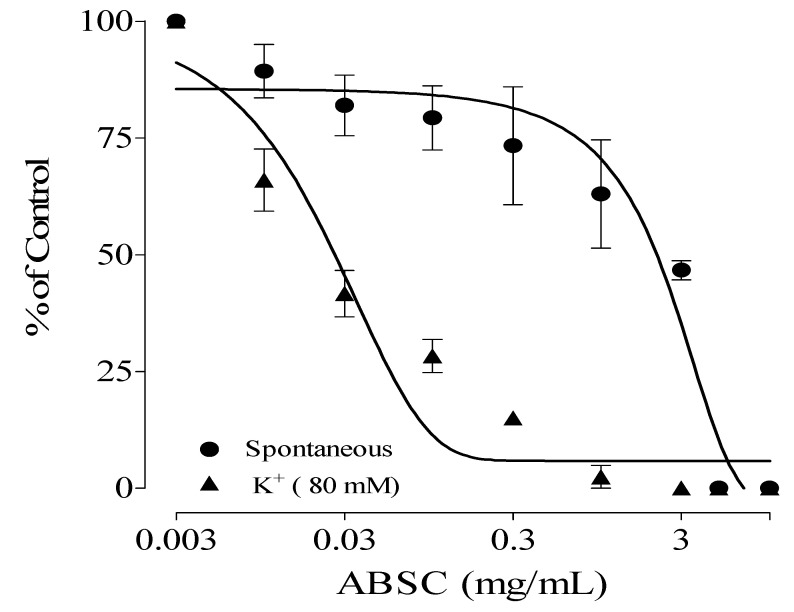
Antispasmodic effects of aqueous extract of *Alstonia boonei* stem back (ABSC).

**Figure 2 molecules-28-07069-f002:**
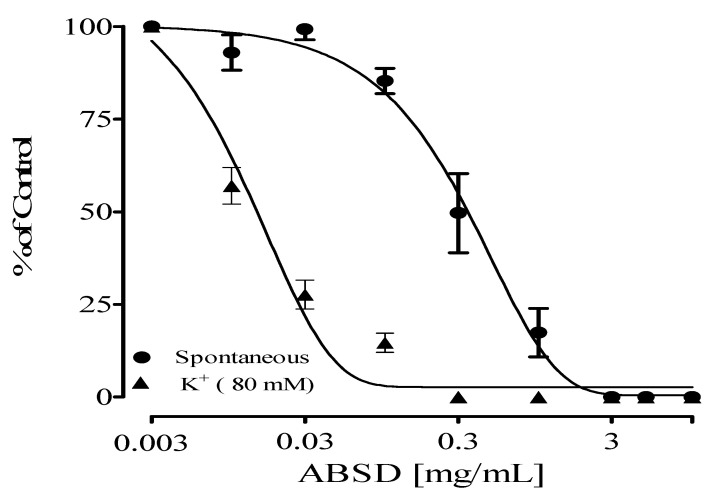
Antispasmodic effects of DCM fraction of *Alstonia boonei* stem back (ABSD).

**Figure 3 molecules-28-07069-f003:**
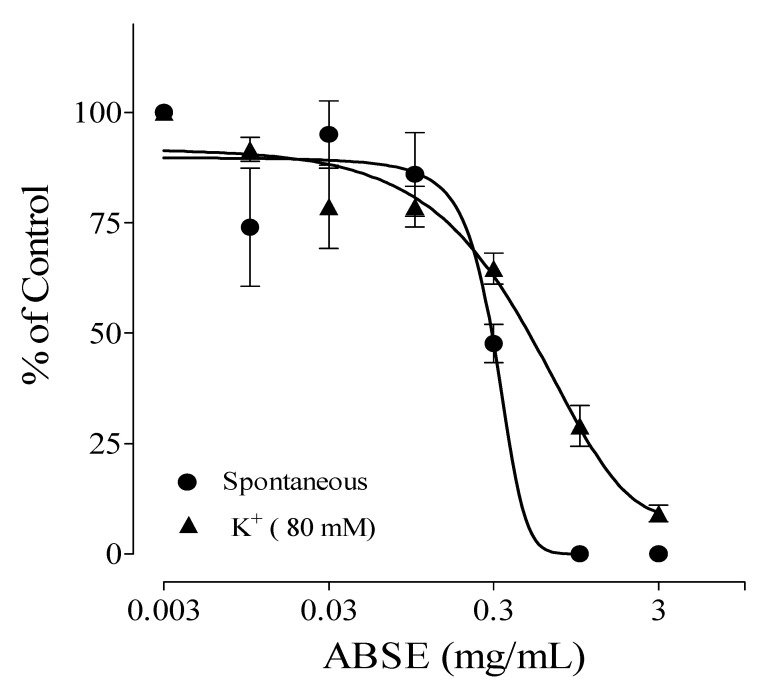
Antispasmodic effects of ethyl acetate fraction of *Alstonia boonei* stem back (ABSE).

**Figure 4 molecules-28-07069-f004:**
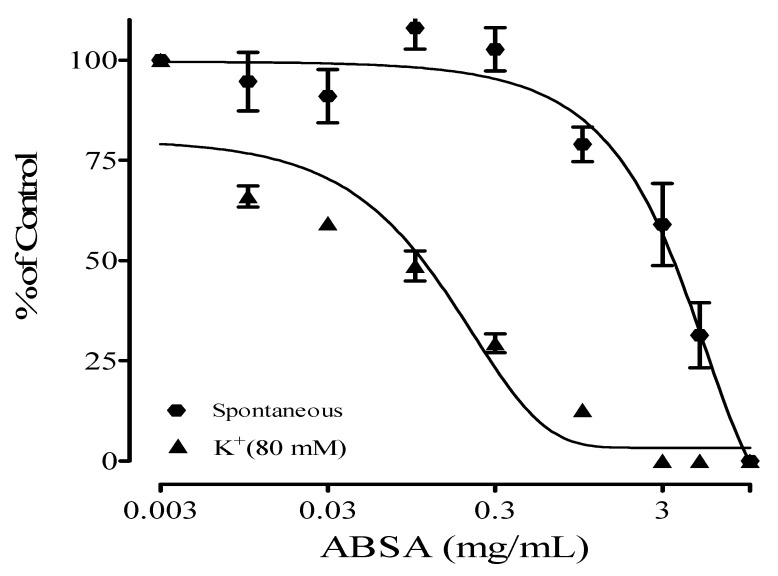
Antispasmodic effects of aqueous fraction of *Alstonia boonei* stem back (ABSA).

**Figure 5 molecules-28-07069-f005:**
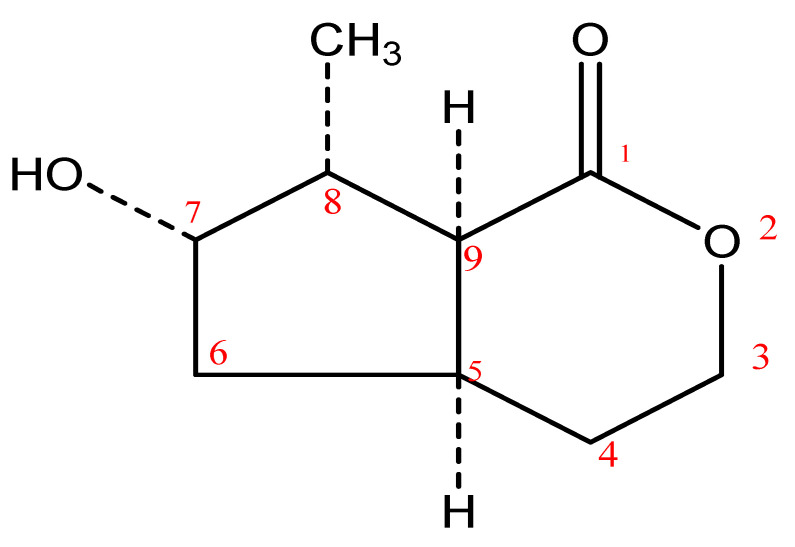
Chemical Structure of Compound **1** (4As, 6S, 7R, 7aS)-6-hydroxy-7-methyl-4,4a,5,6,7,7a-hexahydro-3H-cyclopenta[c] pyran-1-one (boonein). Chemical formula: C_9_H_14_O_3_.

**Figure 6 molecules-28-07069-f006:**
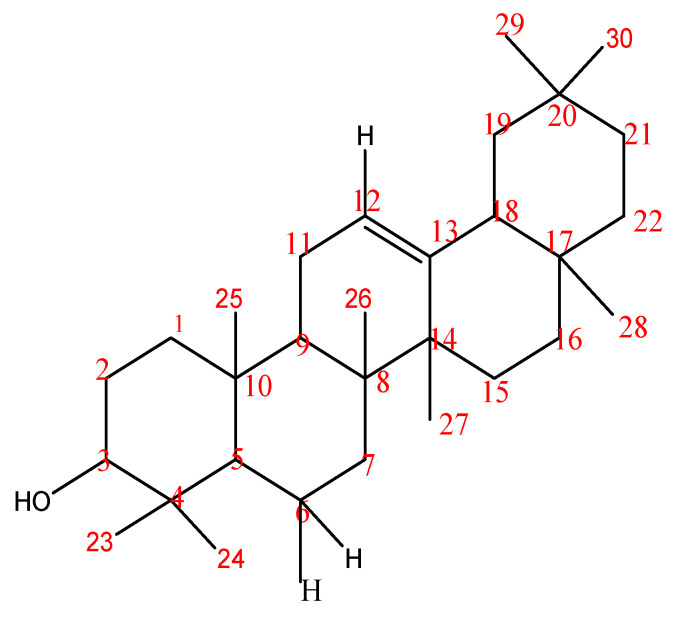
Chemical Structure of Compound **2** (3b-hydroxylolean-12-ene (β-amyrin). Chemical formula: C_30_H_50_O.

**Figure 7 molecules-28-07069-f007:**
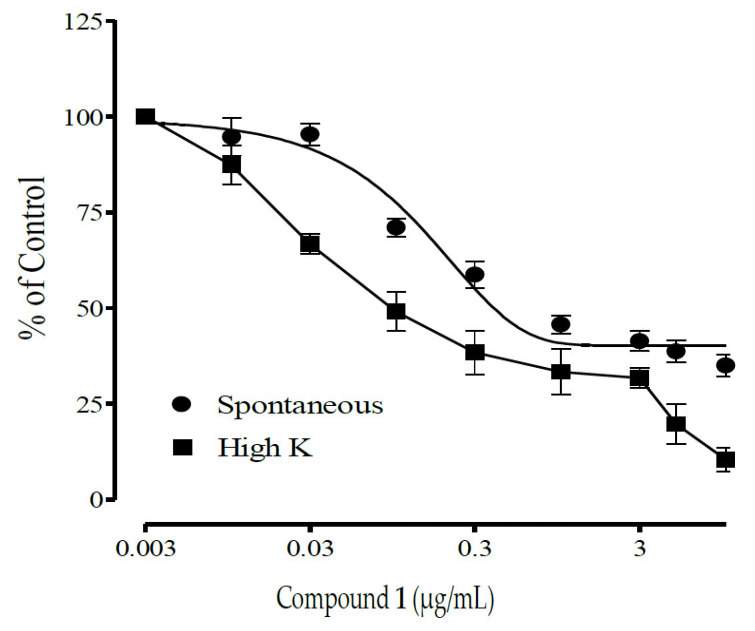
Antispasmodic effects of compound **1** isolated from *Alstonia boonei* stem back on contractions of isolated rat ileum. Compound **1** shows an increase in total relaxation activities on both spontaneous and high-potassium-induced contractions.

**Figure 8 molecules-28-07069-f008:**
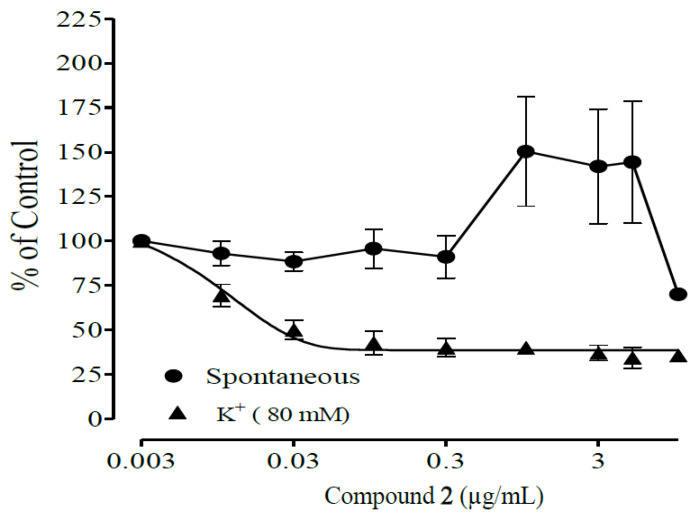
Antispasmodic effects of compound **2** isolated from *Alstonia boonei* stem back on contractions of isolated rat ileum. Compound **2** shows spasmodic (contraction) activities at concentrations 0.003–5 µgmL^−1^ and an antispasmodic (relaxing) effect at 10 µgmL^−1^.

**Figure 9 molecules-28-07069-f009:**
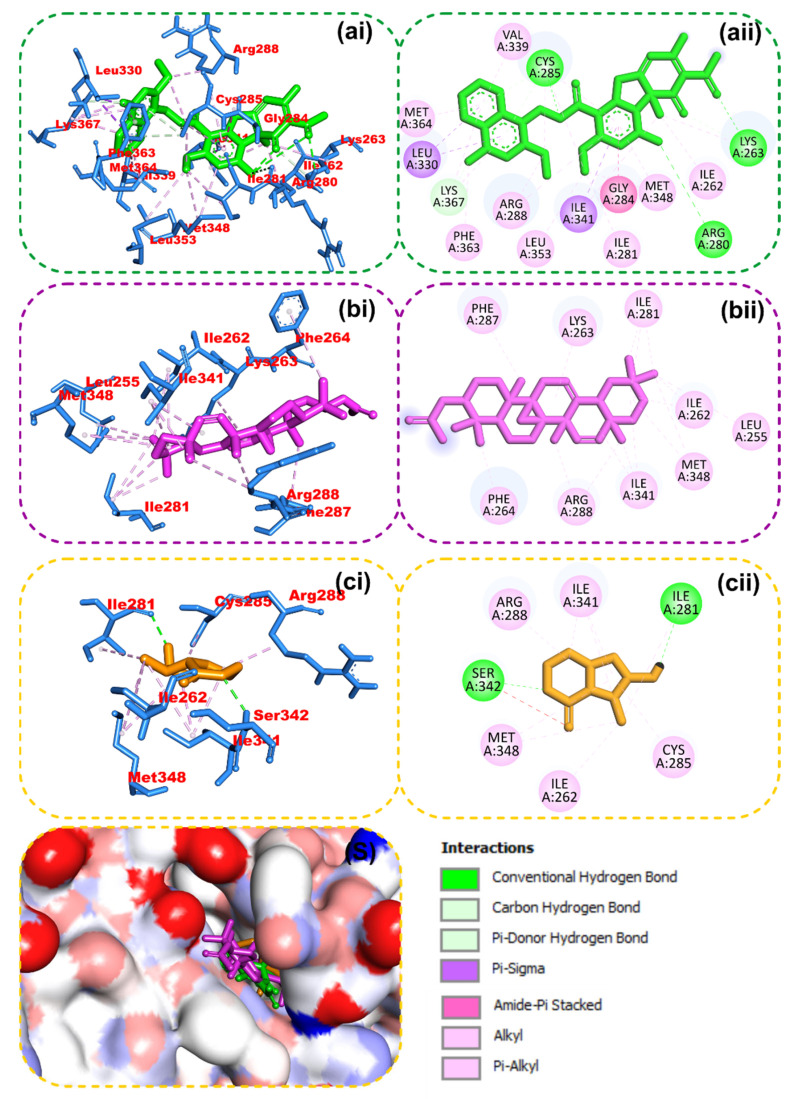
Amino acid interactions of reference inhibitors and isolated compounds from *Alstonia boonei* in the binding site of human PPARgamma-LBD. Stick representations of the ligands are presented in (**i**) 3D and (**ii**) 2D and by colors: (**a**) green: reference inhibitor; (**b**) purple: β-amyrin; (**c**) gold: boonein. (**S**) Surface representation.

**Figure 10 molecules-28-07069-f010:**
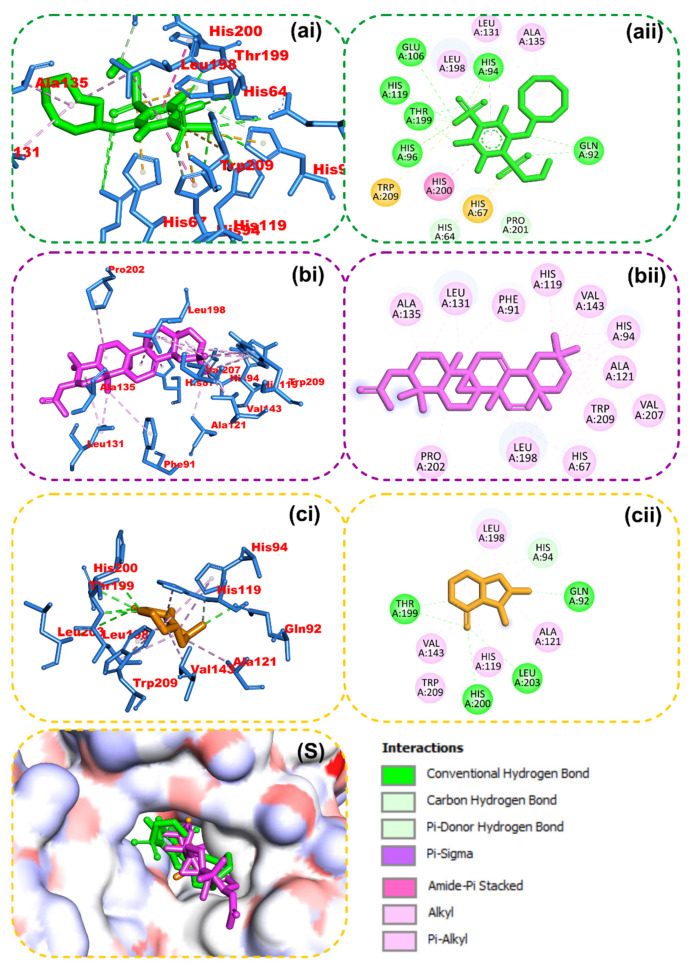
Amino acid interactions of reference inhibitors and isolated compounds from *Alstonia boonei* in the binding site of human carbonic anhydrase isozyme 1. Stick representations of the ligands are presented in (**i**) 3D and (**ii**) 2D and by colors: (**a**) green: reference inhibitor; (**b**) purple: β-amyrin; (**c**) gold: boonein. (**S**) Surface representation.

**Figure 11 molecules-28-07069-f011:**
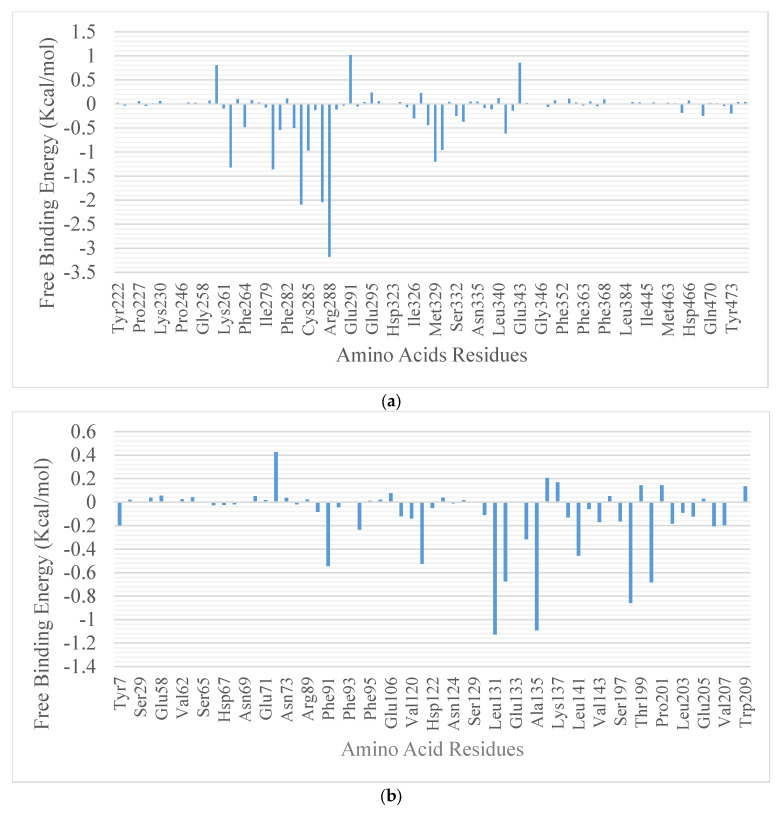
Molecular Mechanics-Generalized Born Surface Area (MM-GBSA) plot of binding free energy contribution per residue within 10 Å of β-amyrin in *(***a**) 4F9M and (**b**) 5E2.

**Table 1 molecules-28-07069-t001:** Effects of crude extract and fractions on isolated rat ileum.

Samples	IC_50_ (mg/mL)	
	Spontaneous	K^+^ (80 mM)
ABSC	1.15 ± 0.10	0.03 ± 0.20
ABSD	0.31 ± 0.02	0.02 ± 0.05
ABSE	0.35 ± 0.03	0.03 ± 0.14
ABSA	2.38 ± 0.65	0.90 ± 0.06

Data are presented as mean ± SEM (n = 3) and evaluated with non-linear regression curve. ABSC: *A. boonei* Stem back aqueous extract; ABSD: *A. boonei* Stem back DCM fraction; ABSE: *A. boonei* Stem back ethyl acetate fraction; ABSA: *A. boonei* Stem back aqueous fraction.

**Table 2 molecules-28-07069-t002:** Effects of compounds **1** and **2** on isolated rat ileum.

Compounds	IC_50_ (µg/mL)	
	Spontaneous	K^+^ (80 mM)
**1**	0.29 ± 0.05	0.09 ± 0.01
**2**	2.20 ± 0.70	0.90 ± 0.10

Data are presented as mean ± SEM (n = 3) and evaluated with non-linear regression curve.

**Table 3 molecules-28-07069-t003:** Molecular interactions of the amino acid residues of human carbonic anhydrase isozyme I (5E2M) and human PPARgamma-LBD (4F9M) with isolated compounds from *A. boonei* and reference inhibitors.

Compounds	Protein	**Hydrogen Bond Distance (Å)**			**Hydrophobic Interaction**
		Numbers	Interacted Residues	Numbers	Interacted Residues
Ref. inhibitor		2	Lys263 Cys285 Arg280	2	Val339 Met364 Leu330 Phe363 Arg288 Leu353 Ile341 Gly284 Ile281 Met348 Ile262
β-amyrin	4F9M	5		3	Phe287 Lys263 Ile281 Ile262 Leu255 Met348 Ile341 Arg288 Phe264
Boonein		4	Ser324 Ile281	8	Arg288 Ile341 Met348(2) Ile262(3) Cys285
Ref. Inhibitor		6	Glu106 His119 Thr199 His96 Gn92	7	Trp209 His69 Leu198 Leu131 Ala135 His67 Pro201 His64
β-amyrin	5E2M			3	Ala135 Leu131 Phe91 His119 Val143 His94 Ala121 Val207 Trp209 His67 Leu198 Pro292
Boonein		4	Thr199 His200 leu203 Gln92		Leu198 Ala121 His119 Val143 Trp209

**Table 4 molecules-28-07069-t004:** Mean and SEM of different energy components of the binding free energy of β-amyrin and boonein to respective proteins.

SYSTEM	ΔVDWAALS	ΔEEL	ΔEGB	ΔESURF	ΔGGAS	ΔGSOLV	ΔTOTAL
4F9M—β AMYRIN	−46.56 ± 3.18	3.66 ± 2.77	21.7 ± 3.39	−5.82 ± 0.48	−42.9 ± 4.18	15.88 ± 3.18	−27.02 ± 2.61
4F9M—BOONEIN	−1.78 ± 3.65	−1.41 ± 4.99	2.62 ± 6.24	−0.29 ± 0.6	−3.19 ± 7.48	2.34 ± 5.79	−0.86 ± 2.33
5E2M—β AMYRIN	−26.86 ± 4.66	−1.24 ± 6.84	18.83 ± 6.46	−3.61 ± 0.66	−28.1 ± 8.07	15.22 ± 6.29	−12.88 ± 4.06
5E2M—BOONEIN	−6.3 ± 6.94	−3.86 ± 7.56	6.89 ± 8.92	−0.92 ± 0.99	−10.17 ± 12.36	5.97 ± 8.19	−4.19 ± 5.22

**Table 5 molecules-28-07069-t005:** The clusters and interactions of carbonic anhydrase isozyme I (5E2M) and human PPARgamma-LBD (4F9M) amino acid residues with boonein and β-amyrin.

	Compound	Salt Bridges			Hydrophobic Interactions		Hydrogen Bonds	
		Cluster Number	Number	Amino Acids	Number	Amino Acids	Number	Amino Acids
4F9M	Boonein	C1	1	K373	0	None	0	None
		C2	2	R397–R443	0	None	2	S394–R397
		C3	0	None	1	V277	0	None
		C4	0	None	2	P304–I409	0	None
	β-amyrin	C1			6	I262–M329–L330–L333–I341–Y473	1	Y473
		C2			5	R280–F287–M329–L330 (2)	0	None
		C3			10	I262–F264 (2)–I281–F287 (2)–I326–L330–L333–I341	0	None
		C4			9	I262 (2)–F264 (2)–R288 (2)–I341 (2)–L465	0	None
5E2M	boonein	C1	0	None	0	None	0	None
		C2	0	None	2	P175–F176	0	None
		C3	1	K159	1	F176	1	N178
		C4	0	None	1	F176	1	T177
	β-amyrin	C1			2	F91–L198		
		C2			5	F91–L131–A135–L141–V207		

## Data Availability

The data presented in this study are available on request from the corresponding author.

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
