# Peer review of "Antispasmodic Effect of Alstonia boonei De Wild. and Its Constituents: Ex Vivo and In Silico Approaches"

_molecules, 2023, doi:10.3390/molecules28207069_

Round 1

Reviewer 1 Report

Dear Editor,

 The manuscript entitled “Antispasmodic Effect of Alstonia boonei De Wild and Its Con- 2 stituents: Ex Vivo and In Silico Approaches “The given information is in this manuscript is useful for researchers and academia and article would be great contribution in related discipline.  The room for improvement is always there and I have suggested some minor revisions. Furthermore, the discussion section is required related citation to improve the discussion. I can extend my services to further review the incorporation of the corrections in article again.

·         Title: reconsider as per your aim and objective - This study aimed to 96 evaluate the antispasmodic effect of Alstonia boonei stem-bark and its constituent

·         L-151- Rewrite-he broad band and DEPT spectra, 149 showed the existence of nine carbon signals including a quaternary carbon, four methine 150 carbons, three methylene carbons and a methyl group. DEPT 90 spectrum showed the 151 existence of four methine (CH) group while DEPT 135 spectrum showed the existence of 152 a methyl group (CH3) whose peak exist upfield (on the positive axis of the spectrum), four 153 methine (CH) whose peaks are found down field (on the same positive axis) w.

·         Abstract- Mention numerical value- .

·         Incomplete sentence- . Alstonia boonei De Wild. (Apocynaceae) sample was collected in September 2015 be- 123 hind the Department of Physiology at the University of Ibadan in Ibadan, Nigeria. Au- 124 thentication was done by Mr. Ifeoluwa Ogunlowo, the herbarium curator at the Ife Her- 125 barium, Department of Botany, Obafemi Awolowo University, Ile-Ife with voucher num- 126 ber FPI 2169

·           Page 2, line 52: The statement “…with potential implications for newer therapeutic targeting [14-16] is incomplete. Kindly complete it.

·          Page 2, line 68: essential oils (caraway and peppermint) should be changed to essential oils (from caraway and peppermint)

·         Page 8, line 214: Samples should be changed to compounds.

·         Figure 11: Please indicate which one is a and b.

·         Page 10, lines 265-267: I will suggest this sentence should be presented as two with appropriate use of punctuation mark.

·          Discussion, line 13: all botanical names should be italicized.

·          Material and method: state the brand of freeze-drier used and how it was used (before or after the use of rotary evaporator)

·         How can you justify that your study is feasible /economical for the field?

·         Please avoid repetition-

·         Please check reference style throughout MS

·         Italic all the scientific names,

·         Remove grammatical mistakes

·         Need to rewrite the conclusion

·         Recheck Legends description is as per figure number and discussion-

·         I urge the authors to improve the English language for better flow of literature

Minor editing of English language required

Author Response

The manuscript entitled “Antispasmodic Effect of Alstonia boonei De Wild and Its Con- 2 stituents: Ex Vivo and In Silico Approaches “The given information is in this manuscript is useful for researchers and academia and article would be great contribution in related discipline.  The room for improvement is always there and I have suggested some minor revisions. Furthermore, the discussion section is required related citation to improve the discussion. I can extend my services to further review the incorporation of the corrections in article again.

  • Title: reconsider as per your aim and objective - This study aimed to 96 evaluate the antispasmodic effect of Alstonia boonei stem-bark and its constituent.

Response: corrected.

  • L-151- Rewrite-he broad band and DEPT spectra, 149 showed the existence of nine carbon signals including a quaternary carbon, four methine 150 carbons, three methylene carbons and a methyl group. DEPT 90 spectrum showed the 151 existence of four methine (CH) group while DEPT 135 spectrum showed the existence of 152 a methyl group (CH3) whose peak exist upfield (on the positive axis of the spectrum), four 153 methine (CH) whose peaks are found down field (on the same positive axis) w.

Response: Tenses corrected.

  • Abstract- Mention numerical value- .

Response: Corrected.

  • Incomplete sentence- . Alstonia boonei De Wild. (Apocynaceae) sample was collected in September 2015 be- 123 hind the Department of Physiology at the University of Ibadan in Ibadan, Nigeria. Au- 124 thentication was done by Mr. Ifeoluwa Ogunlowo, the herbarium curator at the Ife Her- 125 barium, Department of Botany, Obafemi Awolowo University, Ile-Ife with voucher num- 126 ber FPI 2169

Response: corrected.

  • Page 2, line 52: The statement “…with potential implications for newer therapeutic targeting [14-16] is incomplete. Kindly complete it.

Response: corrected

  • Page 2, line 68: essential oils (caraway and peppermint) should be changed to essential oils (from caraway and peppermint)

Response: corrected

  • Page 8, line 214: Samples should be changed to compounds.

Response: corrected.

  • Figure 11: Please indicate which one is a and b.

Response: corrected.

  • Page 10, lines 265-267: I will suggest this sentence should be presented as two with appropriate use of punctuation mark.

Rresponse: Corrected

  • Discussion, line 13: all botanical names should be italicized.

Response: corrected

  • Material and method: state the brand of freeze-drier used and how it was used (before or after the use of rotary evaporator)

Response: freeze-drier (Gunman, Germany)

  • How can you justify that your study is feasible/economical for the field?

Response:  The use of ex vivo approach (which is bio-conservation), and in silico study (to have a clue on the mechanism of the isolated compounds) are economical but not cheap.

  • Please avoid repetition-

Response: corrected

  • Please check reference style throughout MS

Response: all references had been validated

  • Italic all the scientific names,

Response: corrected

  • Remove grammatical mistakes

Response: corrected

  • Need to rewrite the conclusion

Response: corrected

  • Recheck Legends description is as per figure number and discussion-

Response: From the guide to author, the legends for Figures and Tables were presented as results

  • I urge the authors to improve the English language for better flow of literature

Response: corrected

Reviewer 2 Report

In this study, the authors test the antispasmodic effect of the extract of Alstonia boonei stem-bark and its constituents for the first time. The dichloromethane, ethyl acetate and aqueous fractions of the titled species were evaluated for their antispasmodic effect via ex vivo method, respectively. According to 1D, 2D NMR experiments, two compounds (β-amyrin and boonein) were identified from a bioactive dichloromethane extract under the bioassay-guided approach. In this case, both compounds were analyzed for their the antispasmodic activities and relaxed spontaneous and high potassium induced contractions of isolated rat ileum, of which boonein demonstrated greater antispasmodic activity than β-amyrin. The subsequent mode of action was investigated by in silica approach. The experiments were well conducted and the conclusion was reasonably drawn. These findings will benefit the development of natural products-based antispasmodic drugs from the species Alstonia boonei. Thus, it will be recommended to the journal, but a minor revision is required. See below for some suggestions:

1.     Some strongly related literatures have to be added, like Fatima, Memona, et al. Journal of Pharmacognosy and Phytochemistry 2021, 10(2), 121; Yunes, Rosendo A. et al. Planta Medica 1990, 56(2), 242).

2.     The relative configuration of compound 2 should be displayed.

3.     The references (such as 67) should be checked and uniformed according to the guidance of the journal.

4.     The Latin name of the species should be in Italic.

5.     The purity of the isolated compounds should be provided in supplementary material.

 Promote the English language of paper before publication.

Author Response

In this study, the authors test the antispasmodic effect of the extract of Alstonia boonei stem-bark and its constituents for the first time. The dichloromethane, ethyl acetate and aqueous fractions of the titled species were evaluated for their antispasmodic effect via ex vivo method, respectively. According to 1D, 2D NMR experiments, two compounds (β-amyrin and boonein) were identified from a bioactive dichloromethane extract under the bioassay-guided approach. In this case, both compounds were analyzed for their the antispasmodic activities and relaxed spontaneous and high potassium induced contractions of isolated rat ileum, of which boonein demonstrated greater antispasmodic activity than β-amyrin. The subsequent mode of action was investigated by in silica approach. The experiments were well conducted and the conclusion was reasonably drawn. These findings will benefit the development of natural products-based antispasmodic drugs from the species Alstonia boonei. Thus, it will be recommended to the journal, but a minor revision is required. See below for some suggestions:

  1. Some strongly related literatures have to be added, like Fatima, Memona, et al. Journal of Pharmacognosy and Phytochemistry 2021, 10(2), 121; Yunes, Rosendo A. et al. Planta Medica 1990, 56(2), 242).

Response: The first publication is a review while the second publication was published in 1990. We will consider these publications in our research in the future.

  1. The relative configuration of compound 2 should be displayed.

Response: since compound 2 is a well-known compounds, we presented it as it is in the manuscript.

  1. The references (such as 67) should be checked and uniformed according to the guidance of the journal.

Response: The names of all the authors have been included in the citation.

  1. The Latin name of the species should be in Italic.

Response: All latin names and words have been italicized.

  1. The purity of the isolated compounds should be provided in supplementary material.

Response: the NMR spectral for the compounds isolated have been included in the supplementary material.
